# From Child to Adulthood, a Multidisciplinary Approach of Multiple Microdontia Associated with Hypodontia: Case Report Relating a 15 Year-Long Management and Follow-Up

**DOI:** 10.3390/healthcare9091180

**Published:** 2021-09-08

**Authors:** Charlotte Thomas, Frédéric Vaysse, Teva Courset, Karim Nasr, Bruno Courtois, Arnaud L’Homme, Nicolas Chassaing, Alexia Vinel, Isabelle Bailleul-Forestier, Luc Raynaldy, Sara Laurencin-Dalicieux

**Affiliations:** 1Periodontology Department, CHU Toulouse, 31062 Toulouse, France; thomas.ch@chu-toulouse.fr (C.T.); vinel.a@chu-toulouse.fr (A.V.); 2Toulouse School of Dental Medicine, Paul Sabatier University, CEDEX 9, 31062 Toulouse, France; vaysse.f@chu-toulouse.fr (F.V.); tevac@hotmail.fr (T.C.); karnasr@gmail.com (K.N.); brunocourtois.co@gmail.com (B.C.); isabelle.bailleul-forestier@univ-tlse3.fr (I.B.-F.); 3Oral Surgery and Rehabilitation Unit (UCOR), CHU Toulouse, 31062 Toulouse, France; arnaud-lhomme@hotmail.fr (A.L.); raynaldy.l@chu-toulouse.fr (L.R.); 4Paediatric Department, CHU Toulouse, 31062 Toulouse, France; 5Competence and Reference Center for Rare Oral Diseases (CCMR O-Rare), CHU Toulouse, 31062 Toulouse, France; chassaing.n@chu-toulouse.fr; 6Biomaterial Department, CHU Toulouse, 31062 Toulouse, France; 7Oral Surgery Department, CHU Toulouse, 31062 Toulouse, France; 8Department of Medical Genetics, CHU Toulouse, 31059 Toulouse, France; 9Prosthodontics Department, CHU Toulouse, 31062 Toulouse, France

**Keywords:** microdontia, hypodontia, prosthetics, multidisciplinary treatment, oral rehabilitation

## Abstract

Oral rehabilitation of patients presenting multiple microdontia is a real therapeutic challenge. These alterations in size, often associated with other dental anomalies, have aesthetic and functional repercussions for patients and can lead to significant psycho-social consequences. We report here the case of an 11-year-old patient with bilateral sectorial microdontia and agenesis of teeth numbers 13 and 23. She also presented staturo-ponderal delay and a history of acute coronary syndrome with a lower coronary occlusion of unknown aetiology. At first, additive coronoplasties and an orthodontically retained interim prosthesis answered the aesthetic and functional need during childhood and adolescence. Once she reached adulthood, a multidisciplinary meeting was conducted and a treatment plan was established. The decision was made to rehabilitate the upper arch with a permanent bridge and the lower arch with indirect adhesive restorations. This solution solved the problem of the bilateral lateral infraocclusions and tooth agenesis, restoring both aesthetics and function. This paper presents 15 years of management and treatment of a patient presenting multiple microdontia associated with hypodontia. Both the multidisciplinary approach and coordination between the different medical team members was essential to maintain the existing dentition while preparing, planning, and carrying out a personalized treatment plan once maxillofacial growth was complete.

## 1. Introduction

Microdontia is a type of dental anomaly in which teeth are abnormally small. Different types of microdontia exist. They can affect a single tooth or the entire denture and both primary and permanent teeth. Other dental anomalies such as hypodontia may also be associated with microdontia [1]. These alterations in size may be linked to genetic factors and in particular syndromes such as pituitary dwarfism and Rieger’s syndrome. They can also be the result of exposure to environmental factors during dental development such as radiotherapy, chemotherapy, or persistent organic pollutants [2,3,4,5]. Their bio-accumulation can impair enamel, dentin and overall tooth crown and root development [5,6]. Microdontia is real therapeutic challenge for dental surgeons since these patients require long-term dental care (from early interventions during childhood to permanent surgical and prosthetic rehabilitations once growth is complete). These anomalies, depending on their severity, have an important impact and affect the wellbeing and quality of life of patients. They can lead to food blockages, malocclusions, and occlusal discomfort in a static position, thus altering the masticatory function in connection with the lack of wedging [7]. From an aesthetic point of view, they cause numerous discomforts that can have psychological and social repercussions [8,9]. Close collaboration between the different oral medical specialists is essential in order to develop a personalized treatment plan.

Following the CARE guidelines for clinical case reporting, the aim of this paper is to present a 15 year-long follow-up and treatment plan of a young girl presenting multiple microdontia associated with hypodontia. Multidisciplinary approach, management, and communication between the different team members were key factors in its elaboration and success.

## 2. Case Report

An 11-year-old patient came for the first time to the pediatric dentistry consultation of Toulouse’s Teaching Hospital for the management of her multiple microdontia and tooth agenesis. She complained of aesthetic and functional discomfort. Her medical history showed that she also presented staturo-ponderal delay, hypothyroidism treated with levothyroxine (Levothyrox^®^), and a history of acute coronary syndrome with a lower coronary occlusion. Unfortunately, the clinical features did not correspond to a known syndrome. Genetic analysis, including array-CGH (180 K) and whole exome sequencing failed to identify any causal variant. Endo-oral and radiological examinations revealed agenesis of 13 and 23 and lateral infraocclusions related to the presence of sectorial dental microdontia (teeth numbers 15-14-12-22-24-25, according to the World Dental Federation notation system) (Figure 1a,b). Her mouth opening was limited to 23 mm in the premolar region.

Initially, during childhood and because of her mixed dentition, early and preventive care was provided. During that period, additive coronoplasties to reshape teeth numbers 12 and 22 were carried out and an orthodontically retained interim partial fixed prosthesis was made to address the patient’s functional and aesthetic complaints and maintain the available space while facilitating normal completion of tooth and jaw development [10] (Figure 1c,d). Unfortunately, for some time, the patient did not come to the follow-up appointments because of her acute coronary syndrome.

Once she reached adulthood, she was addressed to the Oral Surgery and Rehabilitation Unit (UCOR: Unité de Chirurgie Orale et Réhabilitation) of Toulouse’s Teaching Hospital for oral rehabilitation. Updated endo-oral and radiological examination revealed perfectible oral hygiene, carious lesions under composite restorations, and periapical lesions of endodontic origin (Figure 2a–c). Atypical corono-radicular morphologies of the mandibular premolars with very pronounced occlusal crevices associated with rotations were observed (Figure 2d). Right and left edentulous regions in the maxilla (Figure 2e,f) show insufficient bone volume associated with tooth agenesis and important cortical thickness of the mandibular bone (Figure 2e,f). The maxillary and mandibular 3rd molars also presented microdontia. The patient was still wearing her orthodontically retained interim partial fixed prosthesis and lateral infraocclusions were still present.

## 3. Treatment Plan

This patient’s case was discussed at a multidisciplinary meeting comprising of orthodontists, oral surgery specialists, periodontists, endodontists and prosthodontists in order to establish the most appropriate and long-lasting therapy, considering the medical and psycho-social context. It was decided to first carry out periodontal and endodontic therapy. Depending on the success of theses therapies, and in order to answer the aesthetic and functional discomfort expressed by the patient, a full arch maxilla bridge was chosen as a prosthetic solution. In the mandible, minimally invasive indirect adhesive restorations were recommended to correct lateral infraocclusions and restore the anterior guide.

## 4. Therapeutic Management

After non-surgical periodontal therapy, both teeth numbers 32 and 46 were, respectively, endodontically treated and retreated (Figure 3).

In order to determine the future overall prosthetic project, both a wax-up and a set-up were performed. After aesthetic and functional analysis, the available space for a prosthetic rehabilitation being considered sufficient, an increase in the vertical occlusal dimension was found unnecessary (Figure 4a and Figure 5b,c). A mock-up from the wax-up was performed so that the patient could visualize the prosthetic outcome from an aesthetic, functional and phonetic point of view (Figure 4d). After approval, teeth numbers 12, 22, and 63 were considered non maintainable due to carious lesions, excessive mobility for 12 and 22 and shortness of root for 63 and were extracted. The abutment preparations on teeth numbers 16, 11, 21, and 26 were made using a mock-up to ensure tooth penetration proportion control. Teeth 17 and 27 were kept intact in order to maintain the original vertical occlusal dimension and inter-arch ratio. A first temporary bridge was made from the wax-set-up using self-curing resin (Tab 2000^TM^, Kerr, Ivry-sur-Seine, France) extending from 16 to 26 (Figure 4e). Then, the dental laboratory made a second stronger temporary resin bridge reinforced with a cast metal framework to serve as a reference plane in order to correct the infraocclusions.

Treatment of these lateral mandibular infraocclusions was carried out using overlay-type bonded restorations to allow correction of the Spee curve, resulting in a better distribution of occlusal forces and contacts, restoring masticatory capacity (Figure 5). After sandblasting, the crevices of the premolars were filled with a fluid composite in order to shape the tooth surface prior to restauration (Figure 5a,b). Tooth preparations were done as minimally invasive as possible for overlays on teeth numbers 44, 45, 34, 35, and 36 and an inlay on number 46 (Figure 5c). Due to the small oral aperture of the patient and the number of tooth preparations, an intraoral scanner (Emerald™, Planmeca, Helsinky, Finland) was used to record the different surfaces. This technology also facilitates bite registration. Planmeca’s Plan CAD^®^ Easy software (Planmeca Romexis 5.1.0.R, Helsinky, Finland) was used to design the personalized prosthetic restorations (Figure 5d,e). The parts were then milled in blocks of leucite-enriched glass-ceramic (IPS Empres CAD multi, Ivoclar Vivadent^®^, Schaan, Liechtenstein) or lithium disilicate glass-ceramic (e.max CAD, Ivoclar Vivadent^®^, Schaan, Liechtenstein) depending on the residual thickness, thinner for tooth number 36 (Figure 5f). Finally, they were stained, glazed, and fired. The restorations were bonded, after isolating the teeth with a dental dam, using an adhesive system (Optibond^TM^ XTR, Kerr, Ivry-sur-Seine, France) associated with a dual cure resin composite cement (NX3 Nexus^TM^, Kerr, Ivry-sur-Seine, France) (Figure 5g,h).

Once the lateral infraclusions corrected, the full-ceramic-to-ceramic bridge on zirconia frameworks (Dental Laboratory: Atelier Dentaire, Toulouse, France) was made to replace the temporary one on the maxilla and was extended to teeth numbers 17 and 27, which were prepared using a mock-up to ensure tooth penetration proportion control (Figure 6a). Finally, rehabilitation of the canine-incisor mandibular block was considered. The previously composite restored teeth were infiltrated, also, considering the remaining tooth surface and tooth mobility, indirect adhesive restorations such as Emax veneers (machined lithium disilicate, Ivoclar Vivadent^®^, Schaan, Liechtenstein) on teeth numbers 33 to 43 were made for a more durable rehabilitation and the anterior guide (Figure 6b). The lack of canines and number of missing teeth in the maxilla as opposed to the mandibula associated with the extent of the fixed tooth-supported bridge facing mostly natural teeth resulted in the choice of a group function occlusion.

The final clinical situation answered the patient’s desires, improving masticatory function and aesthetics (Figure 6c–g). Prosthetic, periodontal and occlusal follow-up carried every six months show complete satisfaction from an aesthetic point of view but also functional stability and comfort with the absence of temporomandibular joint pain.

Follow-up at 18 months showed good oral hygiene, complete stability of the oral treatment from a periodontal, clinical and radiological point of view and satisfaction of the patient (Figure 7a–d). Even though there were no functional complaints from the patient, due to the limited amount of bone support in the anterior mandibular region a fixed splint was made to ensure tooth stability.

## 5. Discussion

Pathogenesis and factors responsible for microdontia and hypodontia are still poorly understood. More than 80 different genes and over 150 syndromes have been identified so far as being related to tooth anomalies ranging from tooth agenesis to hypodontia and oligodontia [11]. The genes identified are mostly positive regulators of cell proliferation, negative regulators of cell differentiation, or negative regulators of apoptosis [11,12,13]. This particular patient was the first-born of a non-consanguineous family of four. She had a family history of tooth anomalies. Her father presented an agenesis of a lateral maxillary incisor and her grandmother on her mother’s side a conical central maxillary incisor. Referring to her serious and complicated medical status (staturo-ponderal delay, hypothyroidism, supra aortic stenoses, intracranial aneurysms), she was referred to the competence center for rare oral diseases. No genetic analysis was able, so far, to link it to a known syndrome. To date, whole exome genetic investigations, including array-CGH (180 K) and whole exome sequencing, ruled out various syndromes such as Cockayne and Cohen syndromes.

By causing malocclusions, masticatory disorders as well as aesthetic discomfort, theses dental anomalies bring out numerous aesthetic and functional demands from the patients and their families. They can have an important impact on their quality of life and self-esteem [7,9,14]. Multidisciplinary approaches are required to establish a personalized therapeutic calendar over several years [15,16]. In order to determine a complete and effective treatment plan, diagnosis of these dental anomalies must be carried out at an early stage, preferably in childhood during mixed dentition. Long-term care will include different therapeutic phases, each of which requiring precise objectives. Initially, the early phase aims to temporally respond to the aesthetic and functional demand during the time of growth without carrying out irreversible actions [15]. Orthodontically retained partial fixed prosthesis used in this case is an example of interim prosthesis [10]. It is well accepted by children, often better than removable prosthesis [17]. This device is usually used for aesthetic and functional management during the healing phases of implant treatment procedures [10,18]. It has the advantage of offering a fixed solution without adjacent teeth mutilation and is easy to fit and remove. It is also suitable to restore dentures without excessive mucosal support in order to limit alveolar bone resorption. It can also, if necessary, exert orthodontic forces for the purpose of interception. Then, during adolescence, a multidisciplinary meeting held with the different oral specialists (prosthodontists, oral surgeons, orthodontists, periodontists, etc.) will define the most suitable prosthetic project and articulate the various specialties’ treatment schedules (Figure 8).

Following this meeting and after consultation with the patient, the therapeutic choice was made between of a full arch bridge in the maxilla and indirect bonded restorations in the mandibula. Other options such as those involving implantology are promising and can also be considered as a treatment of choice for patients with dental agenesis [19,20]. However, in these edentulous areas, the bone height and width are very often insufficient and require pre-implant bone assessment and augmentation. Also, in cases of severe hypodontia, the risk of implant failure is higher than in non-compromised patients [21,22]. In our case, the implant solution was not chosen since both extensive vertical and horizontal pre-implant bone augmentations were necessary. The quality of the native bone composed essentially of basal bone was not in favor of extensive grafting due to its poor vascularization and important cortical thickness. Also, taking in consideration the patient’s overall unstable medical history, complex surgery was ruled out in favor of tooth preservation. Finally, two other parameters also guided the prosthetic choice: a perfectible plaque control increasing the risk of long-term implant complications, and a limited buccal opening (23 mm in the premolar region) making it impossible to use an implant drill respecting a proper drilling axis in adequacy with the prosthetic axis. The fixed tooth-supported solution, despite the patient’s age and even though it involved dental preparations, allowed us to harmoniously restore her smile while improving her masticatory function. In view of the different medical and dental parameters, it was, from our point of view, the most suitable solution for this patient [23,24]. The interim temporary restorations made it possible to validate the position of the prosthetic construction, the aesthetic appearance and ensured the absence of temporomandibular joint disorders. Once the occlusal plane was determined and validated by the maxillary interim devices, the indirect adhesive restorations could be made at the mandible. These restored balanced lateral occlusal contacts and the anterior guide, while ensuring the best possible tissue economy. Finally, periodontal and occlusal maintenance were carried every six months. They are essential to ensure the long-term success of this rehabilitation.

Even though the oral treatment lasted for over decade, from the patient’s point of view, restoring her smile increased her self-esteem, and aesthetic and functional comfort. It also had a positive impact on her oral hygiene and compliance since she understood the importance of plaque control in maintaining long term periodontal and oral health prerequisites for long term success of the prosthetic rehabilitation.

To sum up oral rehabilitation of cases comprising of various and numerous dental and skeleto-facial anomalies usually takes place in three phases. The first, during primary dentition, comprises of transitional rehabilitations whose main objectives are functional and aesthetic to accompany craniofacial growth and psycho-emotional development [25,26,27]. The means are essentially orthodontic, restorative using adhesive dentistry techniques and prosthetic, preferably fixed, to promote function [14,24]. At the end of the mixed dentition, when the dental situation is stabilized, a multidisciplinary meeting allows the prosthetic project to be drawn up with the patient and his family. The second phase during adolescence allows for the establishment of favorable conditions for the realization of the future prosthetic project and must emphasize on the multidisciplinary approach of the case [15,28,29,30]. It implements orthodontic and transitional prosthetic solutions. Surgical and/or implant solutions are more likely to be carried out in adulthood [19,31]. The choice of implant placement must be precisely assessed and evaluated for long term mucosal integration knowing that deciduous teeth also show good survival rates [20,32,33,34,35]. Also, particular attention must be paid to aesthetics to promote the psycho-social integration of the patient during this critical developmental phase. Finally, the last phase of prosthetic rehabilitation will be carried out during adulthood according to the personalized prosthetic plan defined. It mainly comprises of implant or more conventional fixed prosthetic rehabilitations. The objectives of which are long-term functional and aesthetic patient rehabilitations [19,22,24,36].

## 6. Conclusions

Oral management of multiple microdontia, often associated with other dental anomalies, is a real therapeutic challenge. It requires a coordinated and experienced multidisciplinary team to maintain the existing dentition while preparing, planning and carrying out a personalized and long-lasting rehabilitation once maxillo-facial growth is complete. This close collaboration is essential in order to answer patient’s complaints during childhood, adolescence and later adulthood and improve their quality of life and overall well-being.

## Figures and Tables

**Figure 1 healthcare-09-01180-f001:**
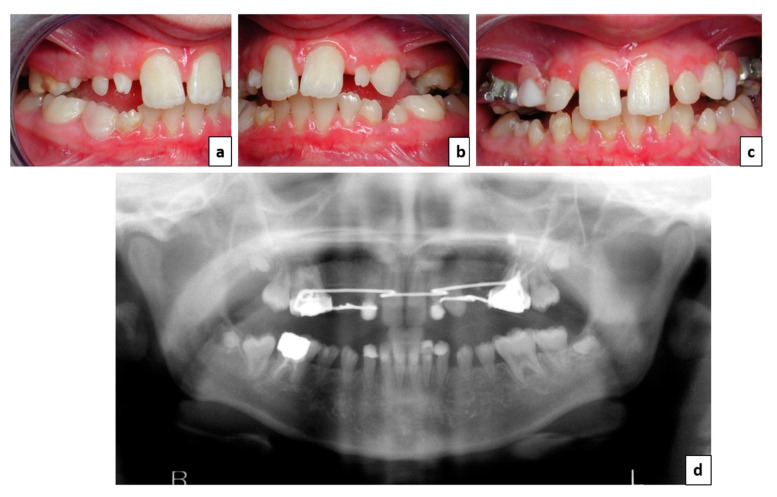
Situation and care provided during childhood. (**a**,**b**) Initial intraoral lateral views; (**c**) endobuccal view after additive coronoplasties and orthodontically retained interim prosthesis; and (**d**) panoramic X-ray of the situation.

**Figure 2 healthcare-09-01180-f002:**
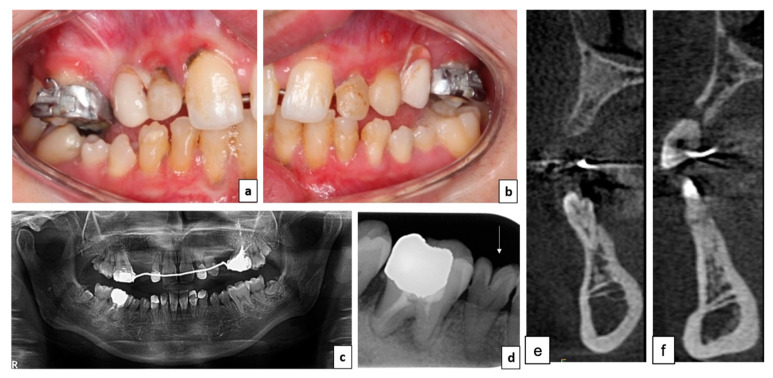
Clinical situation 10 years later (early adulthood). (**a**,**b**) Intraoral lateral views; (**c**) panoramic X-ray of the situation; (**d**) retro-alveolar X-ray of tooth number 45 showing atypical corono-radicular morphology; (**e**) CBCT-scan of the right maxilla canine region; and (**f**) CBCT-scan of the left premolar region.

**Figure 3 healthcare-09-01180-f003:**
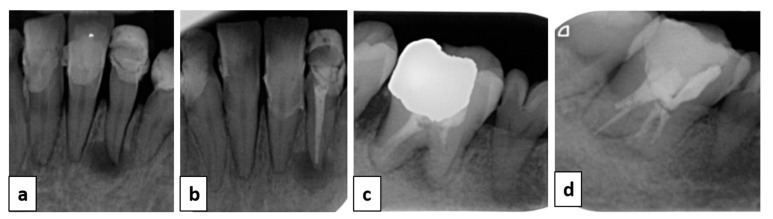
Endodontic treatment of teeth numbers 32 and 46. Retro-alveolar X-rays before and after endodontic therapy of tooth number 32 (**a**,**b**) and number 46 (**c**,**d**).

**Figure 4 healthcare-09-01180-f004:**
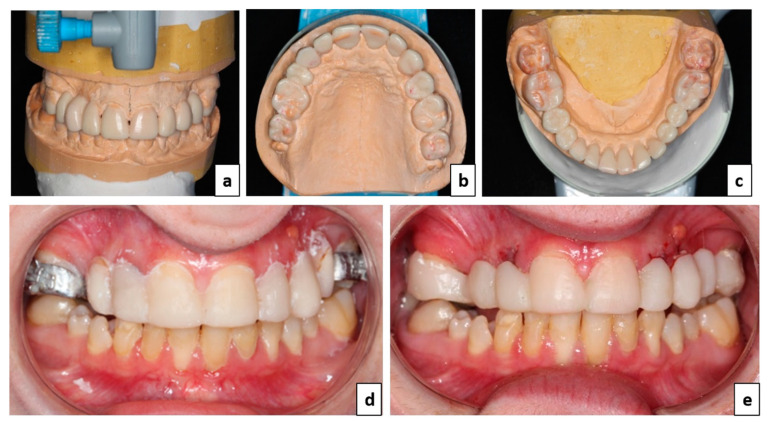
Temporary treatment of the maxilla with a fixed tooth supported prosthesis. (**a**–**c**) Prosthetic project defined by the wax-up and set-up (frontal and occlusal views); (**d**) mock-up from the wax-set-up; and (**e**) 1st generation temporary bridge made from the wax-up and set-up.

**Figure 5 healthcare-09-01180-f005:**
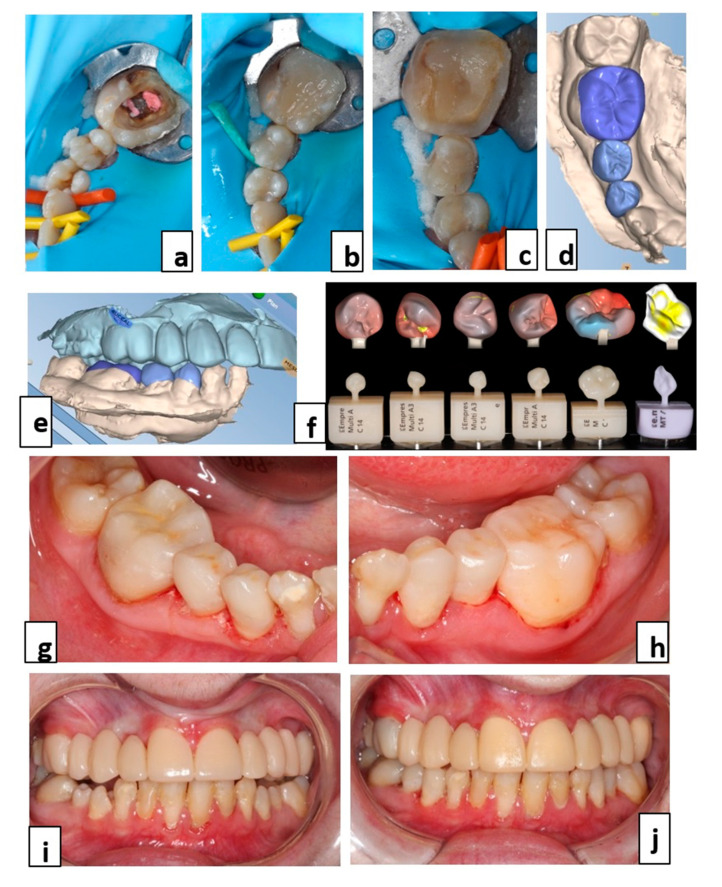
Mandibular rehabilitation in the posterior region and lateral infraocclusion correction. (**a**–**c**) Preparations of the lateral mandibular premolar and molar regions before optical impression; (**d**–**f**) digital modelling of prosthetic parts; and (**g**,**h**) final bonded restorations (intraoral frontal view before (**i**) and after (**j**) correction of lateral infraocclusions).

**Figure 6 healthcare-09-01180-f006:**
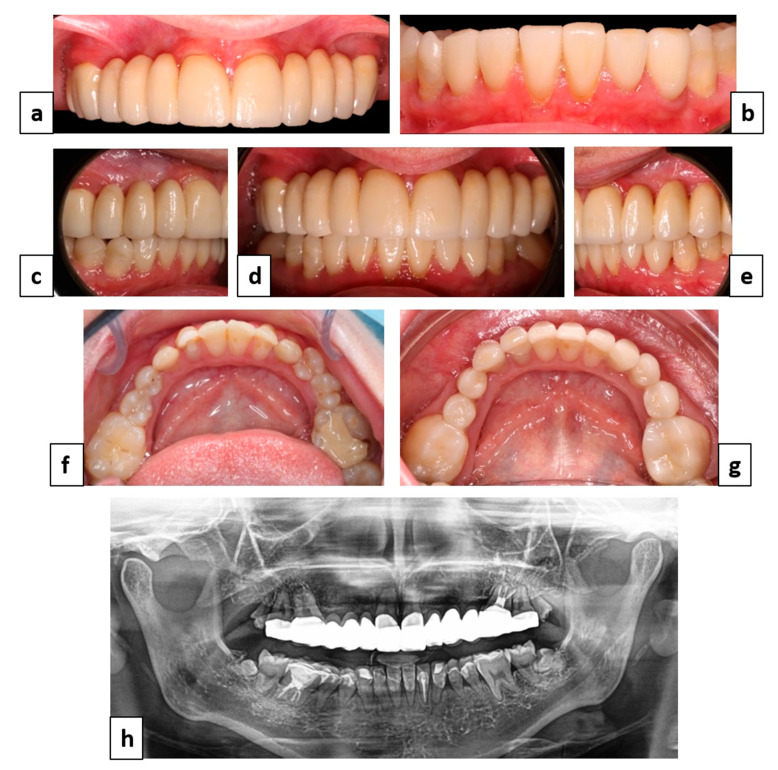
Final maxilla and mandibula rehabilitation. **(a–e)** Final clinical frontal, lateral views; (**f**) initial occlusal view of the mandibular arch; (**g**) final occlusal view of the mandibular arch; and (**h**) final panoramic X-ray.

**Figure 7 healthcare-09-01180-f007:**
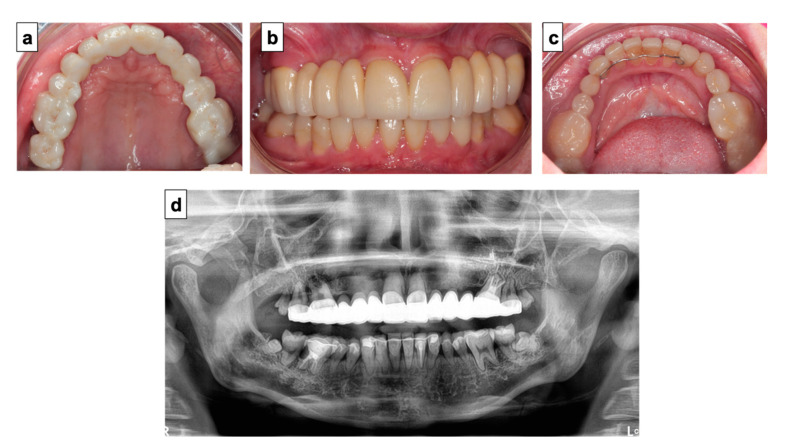
18-month follow-up. (**a**–**d**) 18 month clinical and radiographic follow-up.

**Figure 8 healthcare-09-01180-f008:**
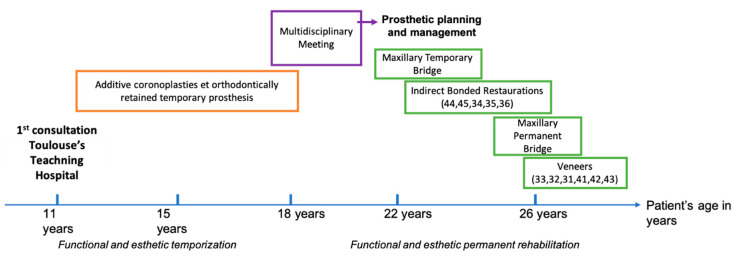
Timeline and summary of management and follow-up over 15 years.

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
