# Peer review of "From Child to Adulthood, a Multidisciplinary Approach of Multiple Microdontia Associated with Hypodontia: Case Report Relating a 15 Year-Long Management and Follow-Up"

_healthcare, 2021, doi:10.3390/healthcare9091180_

Round 1

Reviewer 1 Report

Dear authors, the case is very interesting, due to the misteryous syndrome affecting the poor girl. 

However, being a case report I strongly (and with strongly I mean You have to) suggest to follow the CARE guidelines (

Consensus-based Clinical Case Reporting Guideline Development) to properly report the interest case. 

In addition, to the causes of dental anomalies I would suggest to add the exposure to environmental pollution products ( https://pubmed.ncbi.nlm.nih.gov/32304316/ https://pubmed.ncbi.nlm.nih.gov/25455135/) 

regarding the treatment of the mentioned anomaly at the end I would suggest to sum up, to make a work-flow decision process (literature based of course and so improving the discussion with available literature which I have not seen) so to provide a sort of guide to the clinicians. 

Reviewer 2 Report

This case report testifies to excellent multidisciplinary oral care concerning the functional and aesthetic management of a child with dental anomalies in terms of number and shape, with different stages depending on the age, until a permanent solution. I wanted to congratulate the authors clinically for really putting a smile on the face of this patient.

However, I would like to share some comments in order to improve the manuscript.

1 / On the form, the panel of figure for figure 2 seems incomplete or in inadequacy with the legend and the mansucrit ("Retro alveolar X-rays of tooth number 12, 22 and 63"). In addition, a title for each figure as for figures 1 and 2 would be appreciated.

2/ All the prosthetic steps are well described and perfectly performed. Dental preparations for teeth 17 and 27 could be mentioned.

3/ The choice of a complete maxillary bridge on only 6 dental abutments in this patient with perfectible oral hygiene and periodontal disease ultimately seems the best solution, but somewhat daring. Could you expand on the choice of the type of occlusion in therapeutic management?

4/ Regarding the possible genetic origins of microdontia and hypodontia, could you expand on the state of current research on this subject? An "investigation on the whole genome is still in progress", would it be possible to detail these investigations?

5/ The choice not to resort to implantology was justified by various factors. However, three-dimensional radiological examination sections (Cone Beam-CT or CT-scan) would confirm the insufficient bone volume. In addition, could you justify the following sentence please: “The quality of the native bone was not in favor of grafting due to its poor vascularization”.

Finally, the number of bibliographical references cited on this vast dental subject in terms of genetics, prosthetic rehabilitation, quality of life ... could undoubtedly be more numerous.

Reviewer 3 Report

First of all I would like to thank and congratulate the team that has carried out the treatment of the patient. The effort and follow-up was excellent. However, several questions need to be answered.

- In the evolution of the patient, the stage of bone maturation must be taken into consideration in order to know the growth. Do we have the data of what the stage was before placing the orthodontic appliance? 
- The x-ray shown is from 2007, do you not have the initial x-ray?
- On line 100 in the treatment plan it talks about the specialists who have collaborated in the treatment but does not describe the orthodontists.
- In line 125 it describes the type of material used (Tab 2000TM, Kerr), but does not mention the country, check. The same with the following material references in the rest of the article.
- In the lower arch a lower fixed splint has been placed in the incisor area, explain the reason for this.
- In line 209 he talks about the possibility of placing implants in this case, however he comments that there is not enough bone, but he does not provide radiographic data. It would be convenient to put it as a limitation of the case if he does not have it. A 3D radiological test would be necessary as the last X-ray is not sufficient to assess it.
- The discussion seems to me to be limited and the bibliographical references provided are also limited. In my opinion, it should be improved and the success rates of treatments with implants vs. conventional fixed prostheses should be discussed more extensively.
